# “Talking on the Phone Is Very Cold”—Primary Health Care Nurses’ Approach to Enabling Patient Participation in the Context of Chronic Diseases during the COVID-19 Pandemic

**DOI:** 10.3390/healthcare10122436

**Published:** 2022-12-02

**Authors:** Marcus Heumann, Edurne Zabaleta-del-Olmo, Gundula Röhnsch, Kerstin Hämel

**Affiliations:** 1Department of Health Services Research and Nursing Science, School of Public Health, Bielefeld University, 33615 Bielefeld, Germany; 2Fundació Institut Universitari per a la Recerca a l’Atenció Primària de Salut Jordi Gol I Gurina (IDIAPJGol), 08007 Barcelona, Spain; 3Nursing Department, Faculty of Nursing, University of Girona, 17004 Girona, Spain; 4Primary Care Directorate, Barcelona Regional Management, Institut Català de la Salut, 08006 Barcelona, Spain; 5Division Qualitative Social and Education Research, Department of Education and Psychology, Freie Universität Berlin, 14195 Berlin, Germany

**Keywords:** COVID-19, chronic care, patient involvement, patient participation, person-centered care, primary health care, public health nursing

## Abstract

Strengthening patient participation is considered a crucial element of primary health care (PHC) nurses’ practice when working with chronically ill patients. The COVID-19 pandemic had extraordinary effects on PHC nursing routines and how chronically ill patients’ could be involved in their own care. This study investigates the adaptation of Spanish PHC nurses’ approaches to supporting the participation of patients living with chronic illness during the COVID-19 pandemic. To reach this goal, we interviewed 13 PHC nurses who practiced in PHC centers in Spain. The interviews were analyzed using thematic coding. Three themes emerged from the descriptions of the nurses: (1) High COVID-19-related workload, decreasing health promotion, and chronic care, (2) Emphasis on patients’ and families’ self-responsibility, (3) Expanded digital and telephone communication with fewer in-person consultations. Nurses felt especially challenged to uphold the support for vulnerable groups, such as older people or patients without family support. Future research should focus on how the participation of the most vulnerable chronic patients can be supported in the context of the growing relevance of remote care.

## 1. Introduction

A collaborative environment that enables patient participation is seen as a crucial element in the transformation of health care systems from centering on a biomedical focus to a more holistic biopsychosocial focus [1]. Patient participation is defined as one of the key elements in primary health care (PHC). The importance of participation in PHC has already been highlighted in 1978 by the signatories of the visionary declaration of Alma-Ata [2] and again in the Astana Declaration [3]. Both declarations call for the involvement of individuals, families, and communities in health decision-making processes.

Scholars see PHC as a crucial setting in the prevention and management of chronic diseases [4,5]. Evidence shows that a growing number of patients living with chronic conditions could benefit from greater involvement in their primary care to improve their autonomy and quality of life [6,7,8]. Consistently, chronically ill patients regard their participation in PHC, e.g., their shared decision making and self-management, as an important aspect of their care [7].

Patient participation is a cornerstone of good nursing practice across healthcare settings [9,10,11,12]. In a concept analysis, Nilsson et al. [11] identified instances of patients learning from nurses, reciprocity in negotiations between nurses and patients and the building of caring relationships between nurses and patients as key concepts for patient participation in nursing care. In an integrative review focusing on PHC nursing in the context of chronic diseases, Heumann et al. [10] identified four areas in which nurses intend to promote patient and community participation: (a) establishing a shared understanding of health problems and needs, (b) developing resources and competences for self-management, (c) raising patients’ voices in care planning and service development, and (d) supporting individual and community networks. These participation processes require regular communication between patients and their families and nurses. A variety of factors on an interpersonal, organizational, and systemic level, such as differences in care priorities between patients and nurses or high workloads in primary health care, can make it more difficult for nurses to strengthen patient participation.

The COVID-19 pandemic changed the care environment by exerting robust impacts on the way nurses could promote patient participation in PHC. PHC providers in Europe and elsewhere were forced to transform and adapt their activities to maintain essential (non-COVID-19) healthcare while building up an emergency response to the pandemic [13]. PHC teams needed to react to dynamically changing circumstances and to adapt to constantly changing pandemic-related rules and new information that influenced their working routines [14,15]. A study by Pulido-Fuentes et al. [14] indicated that protective measures designed to prevent the transmission of the coronavirus, such as restricting access to PHC centers and checking patients with COVID-19 symptoms, supplanted prevention and health promotion activities in PHC, which are crucial for strong patient participation. In addition, PHC professionals faced new organizational challenges that resulted from higher workloads due to additional tasks, such as vaccination and COVID-19 testing, staff shortages as a consequence of high COVID-19 infection rates among health professionals, and, especially at the early stage of the pandemic, the insufficient availability of protective equipment [15,16,17,18].

At the same time, technical innovations, such as remote care in the form of remote consultations or patient monitoring via telephone and video conferencing, entered PHC at an unprecedented pace to sustain service delivery and the ability to meet patients’ needs [14,19,20]. However, studies indicate that these solutions have positive, as well as negative, impacts on the quality of care. In a literature review, Silva et al. [20] refer to studies that point out the positive impacts of telehealth consultations. According to these studies, digital communication and telehealth positively impact the establishment of trust and bonds between patients and health professionals, which facilitates patient participation. Furthermore, respect between professionals and patients improved when using these digital solutions. However, there is also evidence of negative impacts on communication, e.g., through the loss of nonverbal communication or insecurity [20].

The COVID-19 crisis has led to extraordinarily rapid transformations in PHC service delivery. Remote care has been one of these major transformations [21]. Those transformations forced nurses to change their way of communicating with patients and families and to strengthen patient participation in everyday life, as well as in care. It is of interest to learn more about how nurses adapted their efforts to support patient participation. It is foreseeable that remote care and other changes in PHC practice will have a lasting influence on the way PHC nurses support patient participation. However, studies that investigate the changes in PHC nurses’ approach to patient participation during the pandemic are scarce. In addition, studies that analyze how these changes affected the relevance of participation for PHC nurses are lacking.

### Aim of the Study and Research Questions

The aim of this study is to investigate how nurses in PHC centers in Spain adapted their approach to support the participation of patients living with chronic illness during the COVID-19 pandemic. To reach this aim, our research questions are: (1) To what extent did PHC nurses experience changes in their practical efforts to strengthen the participation of patients with chronic diseases during the COVID-19 pandemic? (2) What challenges do PHC nurses face when trying to strengthen the participation of chronically ill patients? (3) What strategies do PHC nurses adopt to overcome these challenges? The results of this study can inform the development of person-centered care approaches that build on patient participation and take into account the lessons that were learned during the COVID-19 pandemic.

## 2. Materials and Methods

### 2.1. Study Design

This exploratory study is based on qualitative expert interviews [22] conducted with PHC nurses practicing in Spain. It builds on an international cross-country analysis on PHC nursing in Spain, Germany, and Brazil (for the methods of this study, see Hämel et al. [8]). The interview guideline used in this cross-country analysis was developed prior to the start of the COVID-19 pandemic. However, the data collection process was disrupted by the COVID-19 pandemic and then resumed in the form of Zoom meetings. The data analyzed for this article exclusively comprise PHC nursing during the pandemic, as our analysis indicated that nurses adapted their approach toward strengthening patient participation. With the present study, we aim to investigate these adaptations. In the beginning, we included interviews with practicing PHC nurses in Spain that were conducted between October 2020 and December 2020 (referred to as interview phase 1). The interview guideline consisted of broader questions regarding patient participation in PHC.

To conduct a deeper analysis of these changes, we completed a second phase of data collection from February 2022 until April 2022 that was not included in the cross-country analysis. For this phase, we used an interview guideline that focused explicitly on the changes in and challenges of nurses’ approaches during the COVID-19 pandemic toward strengthening patient participation. The guideline for this study was developed by building on the insights that were gained during the first interview phase.

This study is reported in accordance with the consolidated criteria for reporting qualitative research (COREQ) [23].

### 2.2. Context of the Study: Spanish PHC during the COVID-19 Pandemic

Spain has a PHC-centered healthcare system that offers universal coverage with free access to healthcare for the entire population. It is predominantly financed through taxes [24]. The Spanish PHC system is considered to be one of the strongest in Europe [25]. PHC in Spain has a gatekeeping function, and its services comprise prevention, health promotion, and community care [24]. PHC nurses and physicians work in fixed teams that usually care for the same patient lists [26,27]. PHC nurses in Spain are traditionally responsible for various activities in health promotion, prevention, and chronic care [27]. In recent years, their role in chronic care has expanded, which means that PHC nurses now follow up on chronically ill patients and are authorized to prescribe a limited scope of pharmaceutical products [26].

In Spain, as in other countries, COVID-19 precautions restricted social life and access to services per federal and local government guidelines. Among other measures, two state-of-emergency periods were established between March 2020 and May 2021, lasting approximately three and six months, respectively. They put into place curfews and severe restrictions on mobility and access to different services, including health services. The concrete measures that were undertaken varied between the autonomous communities (“Communidades Autónomas” are the regional states of Spain that have their own legislative and executive rights and duties. They are responsible for offering health care services to their regional populations.). Apart from these measures, quarantine and isolation obligations were in place until March 2022.

The emergence of the COVID-19 pandemic caused profound changes to the organization of PHC in Spain. Many patients were afraid to consult PHC centers because they feared coming into contact with COVID-19. Moreover, health professionals were instructed to minimize physical contact with the population as much as possible. As in many other countries, most in-person consultations in PHC were replaced by remote consultations via telephone and video calls [17,28]. This also applied to the monitoring of chronically ill patients. Most of the prevention and health promotion activities that existed before the pandemic, however, had to be cut back due to pandemic-related restrictions. At the same time, PHC services in Spain provided the first and main point of contact for most patients that had a suspected COVID-19 infection and for the COVID-19 vaccination campaign [17].

### 2.3. Sampling and Field Approach

Respecting the circumstances of the COVID-19 pandemic and the limited availability of practicing PHC nurses during the pandemic, we chose a convenience sampling approach for both phases of interviews. We conducted new interviews until data saturation was reached. Data saturation was characterized by the development of robust categories that could not be significantly developed by adding additional interviews [29]. We recruited study participants from six autonomous communities in Spain to account for regional differences in the organization of the healthcare system, in population structure, and in infection level of the COVID-19 pandemic. In total, we interviewed 13 nurses who practiced in the autonomous communities of Andalusia (n = 1), the Canary Islands (n = 2), Castilla–La Mancha (n = 1), Catalonia (n = 5), Navarre (n = 2), and Valencia (n = 2). It must be noted that the proportion of interviewees from the Autonomous Community of Catalonia is relatively high in this sample. The breakdown of the sample according to sex and qualification is described in Table 1.

Study participants were recruited with the support of local cooperation partners and professional organizations. Some study participants were also recruited through personal contacts from previous interview partners. Potential interviewees were approached via an e-mail that informed them about the goals and the procedure of the interviews.

### 2.4. Data Collection

The first phase of interviews was conducted between October 2020 and December 2020. During this time, seven (n = 7) practicing PHC nurses were interviewed. The second phase of interviews was conducted between February 2022 and April 2022. During this phase, six (n = 6) nurses were interviewed. None of the participants were interviewed repeatedly; however, one interviewee was interviewed as part of the cross-country study before the start of the pandemic. There were no potential interviewees who dropped out of the study after giving their consent to participate. The interviews lasted between 41 and 103 min. As explained, two different interview guidelines were used during the two phases of interviews. Both guidelines were designed according to the episodic interview approach [30,31]. They combined targeted questions with incitements to narrate case examples that occurred in practice. English translations of the guidelines for both interview phases can be found in the Appendix A.

The guideline for the first phase of interviews comprised the following themes: (a) the tasks of nurses regarding primary care and interprofessional collaboration; (b) the promotion of participation of chronic patients; (c) the promotion of participation in group programs and in the community; and (d) the conditions that promote/inhibit the strengthening of client participation as a task of nurses. The questions do not target specific pandemic-related changes or challenges in particular. Those topics were, however, reflected in all of the included interviews either as initiated through interviewees’ narrations or focusing on follow-up questions. The second interview guideline was based on the first guideline, but the questions more specifically targeted the changing practices regarding the promotion of patient participation during the COVID-19 pandemic. It comprised the following themes: (a) the tasks of nurses in primary care during the COVID-19 pandemic; (b) the promotion of the participation of patients with chronic conditions during the COVID-19 pandemic; and (c) the promotion of participation in group programs and in the community.

All interviews were conducted online via Zoom and were audio recorded. All study participants were informed of the goals of the study and the usage of their data. They gave their written informed consent in advance. The interviews were conducted in Spanish using a Spanish language translation of the interview guideline. Interviews in the first phase were conducted by a male postdoctoral researcher, and the interviews in the second phase were conducted by a female doctoral student, neither of whom was an author of this article. The interviewers were not familiar with the interviewees before the data collection. Both interviewers spoke fluent Spanish and were trained in advance by the first author. The first author was present during all the interviews to ask clarifying questions when needed. Both the first author and the interviewers introduced themselves prior to the start of the interviews and explained their interest in the research topic.

The interviews were transcribed in Spanish and later translated into English. The transcriptions and translations were performed by the postdoctoral researcher who conducted the first phase of interviews. To guarantee the protection of the participants’ personal data, personal information, especially names and places, was pseudonymized. The translations were carefully reviewed verbatim by the first author.

### 2.5. Data Analysis

The data analysis for this study was conducted using MAXQDA software based on thematic coding according to Flick [32]. Before coding, all first-phase interviews were carefully reviewed, and all answers referring to the COVID-19 pandemic were selected for inclusion in coding. This was conducted in parallel to the data collection in the second interview phase to decide the number of interviews needed for data saturation [29]. Furthermore, the first author (MH) coded the marked answers from the first phase of interviews and the complete transcripts of the second phase of interviews using open coding, in accordance with the method of Strauss and Corbin [33]. Therefore, the interviews were carefully read and segmented into sections consisting of one or more sentences that were consistent. Those sections were marked with codes following the “W-questions”, according to Flick [32]. The coding structure was continuously reviewed by the third (GR) and fourth (KH) author during the coding process. In the case of disagreements among the authors, the codes were discussed until consensus was reached.

In a second step, the codes were reorganized by the first author in close coordination with the coauthors. They were merged into additional categories composed of sections that constitute a thematic scope and originate in different interviews. This process resulted in three main categories or themes upon which the thematic structure was built (see Section 3).

## 3. Results

During the COVID-19 pandemic, PHC nurses experienced fundamental changes in their approach to promote the participation of patients living with chronic illness in care. We identified the following themes that showed how nurses adapted their approach to support patient participation during the COVID-19 pandemic:(1)High COVID-19-related workload and a decrease in health promotion and chronic care;(2)Stronger emphasis on patients’ and families’ self-responsibility;(3)Expanding digital and telephone communication and fewer in-person consultations.

### 3.1. High COVID-19-Related Workload, Decreasing Health Promotion, and Chronic Care

According to all interviewed nurses, dealing with the pandemic required refocusing a considerable amount of nursing care on more ‘urgent’ and demanding COVID-19-related activities, such as testing, vaccination or following up on patients with (mild) COVID-19. In contrast, many well-developed activities regarding health promotion for persons with chronic diseases, particularly group and community work, such as the establishment of groups for patients with diabetes, have come more or less to a standstill. One nurse summarized: “*Prevention activities and health promotion activities have stopped being carried out in greater numbers*” (Phase 2 I1, 8) (In accordance with the COREQ guidelines [23], we present translated quotations from our interviews to illustrate the themes. These quotations are presented in italic font and are clearly identified by phase, interview number, and passage number in the transcript.). In some cases, this standstill continued after pandemic-related formal restrictions on group meetings were lifted because COVID-19-related activities were still restricting the resources that were needed to continue group and community activities.

Although open forums or proactive initiatives that the PHC centers offered for patient education, counseling, and health promotion through groups came to a standstill, some interviewees tried to counter this by integrating health promotion activities into individual consultations. However, the additional time needed for individually tailored information was also lacking. Moreover, some nurses decry that the benefit of group activities lies in supporting patients’ engagement with peer groups and cannot be replaced solely by nurse–patient encounters.

*“It is true that working with health education in groups works better than doing it at the individual level because we can generate synergies between participants that allow people to engage and participate deeply in their own self-care”*.(Phase 2 I3, 23)

The nurses stated that “*the monitoring of chronic pathologies is especially quite troublesome*” (Phase 1 I12, 12). On the one hand, this was explained as a problem caused by some patients withdrawing from regular healthcare visits in fear of a COVID-19 infection: “*[…] fear that the virus is in the health center, and they are going to get sick*” (Phase 2 I2, 30). On the other hand, nurses also reported that some patients “*have asked us if we could visit them more often*” (Phase 1 I14, 26) after nurses limited their home visits because they were not able to conduct self-management activities on their own.

*“We do more tests for COVID-19, more PCR, we have to follow up with patients with COVID-19. And, then of course, the day has only 24 h. The work shift does not involve so many activities. In the end, what is left undone? Surely, the tasks related to more personalized primary care for complex chronic patients”*.(Phase 1 I10, 36)

Consequently, and in particular during times with high COVID-19 case numbers in the respective communities, nurses were not able to maintain relationships with chronically ill patients and their families as closely as they previously had done. In their eyes, an important element of their approach to supporting patient participation was thus weakened. One nurse explained that:

*“Well, it’s true that they didn’t come, because everything was also ‘armored’ […]. We dedicate ourselves exclusively to the care of patients infected with COVID-19. Therefore, when we resumed face-to-face activities, the patients were in critical condition”*.(Phase 2 I6, 7)

Even those nurses who were able to continue the basic monitoring of chronically ill patients (mostly via telephone) stated that they ‘failed’ because they realized that some of ‘their’ patients had been “*left unattended*” (Phase 2 I2, 24). Nurses, for example, realized that without their support, patients were often not able to conduct health-promoting behavior on their own: “*the elderly have lost a lot of mobility, that the blood glucose levels are higher. Why? Patients are exercising less. Older people are experiencing cognitive decline due to fewer personal relationships*” (Phase 1 E10, 18). Most of the nurses felt responsible for the worsening of chronic patients’ health status because they were unable to meet the demands that they had set themselves for enabling patient participation and supporting health promotion and self-management.

The nurses also related that they prioritized consultations with chronically ill people. They stressed that they continued home visits for at least the most vulnerable chronically ill patients, especially those in palliative care and the ones that lacked family support. During these consultations, however, they focused their support more on basic care activities, such as wound care and medication. At the same time, to prevent precarious care situations, they tried to teach patients and family caregivers how to deal with care needs on their own.

### 3.2. Emphasis on Patients’ and Families’ Self-Responsibility

Most interviewed nurses realized that the self-responsibility of chronically ill patients and family caregivers became more important during the pandemic. As physicians and nurses were not able to see and examine patients with chronic diseases with their accustomed regularity, “*the patient or his family had an important responsibility in monitoring symptoms to see if they had worsened or not*” (Phase 2 I1, 13). This also meant that nurses increasingly relied on patients’ and caregivers’ ability to notice and identify an acute situation that required medical or nursing care. For those nurses, patient involvement changed from being something that they supported to something they had to assume, even though they knew that patients might have problems realizing this.

*“Well, there were basically no other option here. Before the pandemic, we tried to get people involved, but it was more complicated. However, precisely because of the pandemic, with the mantle that one appeared at a patient’s house, disguised as an astronaut, they immediately came to their senses”*.(Phase 2 I4, 21)

The nurses continued to educate patients and families on how to practically conduct necessary care tasks and informed them of the warning signs that indicate the necessity of medical or nursing consultation: “*We explained the signs of decompensation of the pathology of each person and the cases in which they had to notify us*” (Phase 2 I5, 17). However, after having ‘sufficiently’ informed patients of the satisfaction of the nurses, some of them changed their approach from only offering to support patients’ and families’ participation to demanding this participation by telling them “*[…] This is what I can do. I can control these problems here. However, (some things) you are going to do by yourselves*” (Phase 2 I3, 29). By calling patients on a regular basis to “*[…] update some information, to see how they were doing*” (Phase 2 I2, 24), nurses tried to encourage patients’ and families’ participation. In this context, the nurses found it important to assure patients and caregivers of their ability to master these challenges on their own: “*[I] congratulated them [the patients] because they were doing a good job. Even if it wasn’t perfect, they were getting involved*” (Phase 2 I2, 26).

Not all patients with chronic conditions could act in a self-responsible and self-managed manner. Nurses observed that conducting complex care activities on their own and detecting warning signs were overwhelming responsibilities, especially for patients living alone or for those who rely on others’ support due to their poor health status. However, some nurses stated that because of the limited contact they had with patients, they were not always able to notice when patients and families had problems following instructions. In some cases, they failed to detect critical strain that chronically ill patients and caregivers experienced in attempting to conduct nursing or medical self-care. One nurse explained the problems that occur when monitoring patients through remote care rather than by a physical examination:

*“And, in this process, (a lot of information) was lost (…) and there are also things that sometimes patients don’t tell you, but you see with an electro. When you have a naked patient in front of you (…) if he is dirty, you see it”*.(Phase 2 I6, 17)

### 3.3. Expanded Digital and Telephone Communication with Fewer in-Person Consultations

All interviewed nurses stated that the expansion of remote consultations, instant messaging, and e-mails in patient communication was one of the major changes in PHC practice during the COVID-19 pandemic. It also had a strong impact on their approach to strengthening the participation of patients with chronic diseases and that of their families in primary care.

When PHC nurses had to reduce physical contact and infection risks, “*what happened was an acceleration of education related to self-care through other means*” (Phase 2 I5, 8-9). Nurses provided education and self-management support through a combination of telephone calls and a variety of digital tools, such as video conferencing, e-mail, or instant messaging. Some nurses stressed that in this context, by being able to reach ‘their nurse’ directly in multiple ways, patients and their families had more control over when and to what extent they wanted to be supported during their care. In addition, nurses were able to adapt their education activities to the needs of the patient group they were caring for. For example, they had to “*[…] carry out the whole training process by videos because patients could see it over and over again*” (Phase 2 I4, 29) or record podcasts to reach younger patients.

However, the nurses were worried that the growing use of digital communication tools could be a barrier for particular patient groups: “*However, it is true that the poorest patients or those with less access to the internet or older patients had faced slightly higher difficulties”* (Phase 1 13, 22). Those patient groups who had problems accessing digital communication, especially patients with dementia or those in palliative care, were, according to some nurses, the ones who particularly needed individualized support and targeted education to be able to participate more strongly in their care. Nurses tried to compensate for these differences in access by using more telephone calls and by prioritizing those groups when conducting home visits and face-to-face consultations (see Section 3.1).

It was important for nurses to maintain basic communication with chronically ill patients and their caregivers. They tried to give patients and families confidence in their heightened involvement in their care process. Most nurses did so by continuously ensuring that they ‘are there for’ the patients and families when needed. They tried to be more easily accessible than they were prior to the COVID-19 pandemic. However, this rise in telephone or messaging communication was demanding for nurses as they had to reorganize their working routine in a way that allowed them to quickly respond to patients’ and families’ questions and concerns. In addition, some nurses saw it as important to keep a ‘professional’ distance from patients and families.

*“And I did try to keep that distance, because if I give them my personal phone number, I would work all day long and it doesn’t suit anyone. Neither the patient nor I. Therefore, patients can use email—in case they want to send something in writing or some image. If not, they have the telephone. I use the phone a lot. I talk a lot on the phone with the patients to see how they are doing”*.(Phase 2 I2, 32)

The lack of nonverbal communication when communicating over the telephone or by digital tools was also seen as problematic by nurses. PHC nurses reported that they were often not able to detect psychosocial problems that hinder chronically ill patients’ or their families’ active involvement in care, stating “*[…] in fact talking over the phone is very cold*” (Phase 1 E10, 18). Consequently, when communicating with chronically ill patients and their families, especially with those they consider the most vulnerable, nurses also focus on nondisease-related problems, such as social isolation or mobility. In addition to technical nursing or care support, some nurses also tried to provide emotional support for patients and families who were struggling with their situation during the pandemic.

*“During the pandemic, we called them [the patients], and they told us about their experiences: ‘I have measured my blood pressure, and everything is fine. What I have is a lot of fear. What I have is that I am very lonely and nobody brings me (…) ‘. There have been consultations full of emotion. Not only in terms of their diseases, because, in truth, (the diseases) have moved to second place”*.(Phase 1 I15, 15)

## 4. Discussion

As a result of this study, we identified three approaches experienced by PHC nurses in Spain that show how they adapted their support for the participation of patients living with chronic illnesses during the COVID-19 pandemic. (1) Nurses decreased health promotion and chronic care activities in reaction to high COVID-19-related workloads and restrictions; they argue, for example, that they were no longer able to continue regular follow ups for chronically ill patients. (2) To adapt to the situation, nurses stressed that the self-responsibility of patients and families was more relevant during the pandemic; furthermore, nurses pronounced that they had to ‘rely’ on the ability of patients and families to self-manage their conditions. In this context, however, nurses worried about vulnerable patient groups who might not be able to self-manage sufficiently. (3) Nurses relied on telephone and digital communication, as in-person consultations had to be kept to a minimum. They reflected that the lack of nonverbal communication decreased their ability to strengthen the psychosocial health of chronically ill patients.

### 4.1. Changes in the Practical Efforts of Nurses to Strengthen the Participation of Chronically Ill Patients

Our results show how nurses felt forced to adapt their efforts to strengthen the participation of chronically ill patients and their families in their care. Many studies provide evidence that PHC professionals had to cut back routine checkups and follow-up visits for chronically ill patients [34,35,36]. The PHC nurses in our study reported that they concentrated their activities in chronic care and self-management support on issues that were mostly related to elementary nursing, such as wound care and medication management. Meanwhile, they feared the consequences of this changed approach. They attributed a lack of psychosocial support to the growing trend of remote care by digital and telephone communication as they considered this to be an ‘unemotional’ encounter. In line with these observations, chronically ill patients reported less psychosocial support during the pandemic in other studies [37,38].

The increasing prevalence of remote care and the use of digital communication tools fundamentally changed PHC nurses’ working routines (see also [14,20]). In accordance with the studies that Silva et al. [20]. highlight in their review, the nurses in our study see a loss of nonverbal and face-to-face communication as counteracting their efforts to strengthen patient participation. In particular, nurses claim that the participation of (especially vulnerable) patient groups is triggered by their (limited) access or capability to taking part in remote consultations or to using digital communication tools. PHC professionals in other studies echo this concern, as they fear that older and socially disadvantaged patients might not be able to make use of remote care and digital communication tools [39,40,41,42]. In this context, a literature review by Matenge et al. [36] presents evidence that issues of access to technology or low technology literacy among some patients challenged the implementation of remote care.

### 4.2. Challenges That PHC Nurses Face during the COVID-19 Pandemic

The results of multiple studies, including ours, indicate that, particularly in times of high case numbers of COVID-19, it is challenging for PHC nurses to fulfill new COVID-19-related duties, such as testing and vaccination, while simultaneously continuing to conduct routine tasks, such as monitoring chronically ill patients [43,44,45]. As in our study, Pulido-Fuentes et al. [14] showed that nurses and other health PHC professionals in Spain felt bad for having to cut back on activities regarding health promotion and prevention. In this context, some nurses we interviewed for our study pointed out that they found it ethically problematic to emphasize or demand patients’ self-responsibility during the pandemic. Studies in the hospital setting found that nurses were pressured by ethical dilemmas and moral distress when caring for patients during the COVID-19 pandemic [46,47]. Studies investigating those issues for primary health care nurses are scarce. Further studies should also address the issue of ethical dilemmas for nurses when working toward strengthening patients’ participation. There is also a need for research on how nurses can deal with these dilemmas.

The participants in our study also felt challenged by the expansion of accelerated communication demands from their patients through open digital communication tools, such as instant messaging, numerous emails, and (unanswered) telephone calls. They found it increasingly hard to set boundaries between their private and professional life as they wanted to be reachable for ‘their’ patients as much as possible but also needed to protect their own well-being and health, which is a distress also described by Pulido-Fuentes et al. [14] for PHC nurses and other PHC professionals. Mohammed and colleagues’ [41] study consistently refers to PHC professionals’ fear of patients overusing virtual services. However, this critique of patient behavior was more prevalent among GPs (36.8%) than among PHC nurses (16.2%).

### 4.3. Strategies That PHC Nurses Adopt to Overcome These Challenges

PHC nurses who were interviewed in our study tested their own strategies for strengthening chronically ill patients’ participation despite the challenges presented by the COVID-19 pandemic:(1)PHC nurses tried to integrate health promotion activities, which were formerly conducted mainly in groups, into individual (virtual) consultations, e.g., by teaching patients how to establish health promotive behaviors. Nurses also reminded patients that they ‘are there for them’ and offered their help when needed.(2)PHC nurses prioritized vulnerable patient groups, such as older people, chronically ill patients living alone, or patients in palliative care, when they conducted limited numbers of home visits and in-person consultations. It was important for nurses to offer continuous support for these patient groups despite the COVID-19-related restrictions because most of these patients were not able to self-manage and conduct care activities on their own, as other groups were. This strategy aligns with study results from the UK that found that the number of PHC consultations with older people and children was reduced to a lesser amount than those with other age groups [48] and with the results of a literature review by Kumpunen et al. [13].(3)To encourage and support patient participation, nurses found new, innovative ways to train patients and to communicate with them through new techniques, such as recording videos with their phones or creating podcasts to explain self-management strategies to patients. While nurses described digital communication tools as being the only measure with which they could continue communicating with patients at the onset of the pandemic, they started to combine remote care and face-to-face visits as the pandemic continued. Future research also needs to investigate how the challenges uncovered in this study can be overcome by nurses in approaches that combine remote and presential care.

### 4.4. Theoretical Considerations

In nursing, the essence of patient participation has been seen to be foundationally based on developing and maintaining a trusting relationship between nurses and patients [9,10,11,12]. These conceptual approaches to patient participation in nursing care, however, consistently assume that repeated direct personal contact between nurses and patients is possible. In the model cases that both Nilsson et al. [11] and Sahlsten et al. [12] developed as part of their concept analyses of patient participation, a nurse–patient relationship develops after a respective patient has had a face-to-face meeting with a nurse that caused them to feel appreciation and feeling that they had been heard by the nurse. Nilsson et al. [11] describe this as a patient and nurse that “get along well” [11] (p. 248). Heumann et al. [10] found that nurses establish collaborative relationships with patients and families when trying to strengthen their participation. The results of this study, however, highlight how relationship building was challenged by the predominance of online and telephone consultations characterized by a lack of nonverbal communication and lack of personal visits during the COVID-19 pandemic. As it is conceivable that remote care will remain integrated into regular (primary) care in the future, further studies should include deliberations on how relationship building can proceed despite limited personal contacts between nurses and patients and on the ways that remote care and digital communication influence nurses’ approaches toward patient participation. In this respect, the findings of our study indicate that the situation of vulnerable groups, such as older people, people with low incomes, or those in palliative care, needs to be carefully considered with respect to their special needs.

### 4.5. Strengths and Limitations

Our interviews were conducted in two phases, the first occurring in late 2020, ca. half a year into the pandemic, and the second one being conducted in the first quarter of 2022, ca. two years after the beginning of the pandemic. We could, therefore, account for changing conditions over the course of the pandemic, such as changing incidence rates and a (partial) relaxation of restrictions due to strong vaccination dissemination in Spain. However, the fact that two different guidelines were used during the two phases of interviews could be seen as a limitation. In addition, the fact that the guideline for the first phase did not contain explicit questions about pandemic-related changes needs to be listed as a limitation. We addressed these aspects by building the guideline for the second interview phase on the knowledge that we gathered during the first interview phase while we also followed the structure and content of the first interview guideline to retrieve comparable data and enable data integration of both interview phases.

The goal of our qualitative study was to analyze and present experiences that make up close to everyday life knowledge about the circumstances under which nurses supported patient participation during the COVID-19 pandemic. We did not have the goal to gather representative data about the strategies that nurses use to strengthen patient participation during the COVID-19 pandemic. Against this background we did not aim for a high number of interviewees and representativity but for data saturation. To achieve this, we involved interview partners with a high variation concerning different parameters: We included PHC nurses from six autonomous communities. The nurses we interviewed worked in rural and urban settings on the Spanish mainland and on the remotely located Canary Islands. We, therefore, were able to consider the different conditions and different courses of the pandemic in these respective areas. In this context, however, it needs to be noted that the autonomous community of Catalonia is more strongly represented than the others and that more nurses practiced in urban areas than in rural areas.

The translations of transcripts could also be considered a potential source of bias in qualitative research [49]. To reduce the risk of bias and to minimize differences in wording following translation, the interviews in this study were transcribed and translated by the same person, who also conducted part of the interviews. In addition, the transcripts were carefully red and checked by the first author, who has intermediate Spanish skills. Ambiguities in translation were discussed between the first author and the translating person. However, the risk of bias cannot be totally eliminated by these measures.

## 5. Conclusions

PHC nurses’ approach to supporting the participation of patients living with chronic illness in their own care changed fundamentally during the COVID-19 pandemic. Nurses faced high COVID-19-related workloads and had to reduce physical contact with patients. Consequently, they constrained health promotion activities and the monitoring of chronically ill patients and emphasized patients’ and family’s self-responsibility for self-management and care. Support for the participation of chronically ill people during the COVID-19 pandemic was characterized by remote care and digital or telephone communication, which forced nurses to adapt their work routines and communication strategies. Future research should particularly focus on how the adaptation of remote care affects the participation of the most vulnerable among chronic patients to take part in remote care activities. Now is the time to enable nurses to develop approaches to strengthen the participation of vulnerable groups despite the growing relevance of remote care and potential future pandemics. These approaches should also consider that patient consultations in the future will partly be conducted as remote consultations, even though pandemic-related restrictions have been lifted.

## Figures and Tables

**Table 1 healthcare-10-02436-t001:** Study sample.

Interview Phase	Sex		Qualification		
	Female	Male	Graduate Level	Specialization in Family and Community Nursing	Master’s Degree or Higher
1	5	2	4	1	2
2	5	1	0	4	2
Total	10	3	4	5 *	4 *

* Two nurses, one in each interview phase, were qualified with a specialization in family and community nursing, as well as having a master’s degree. They are included in this list as “Master’s Degree or Higher”.

## Data Availability

Given the potentially disclosive nature, entire interview transcripts will not be made publicly available. They will be deposited at Bielefeld University, and reasonable requests for secure research access will be considered. Please contact: mheumann@uni-bielefeld.de.

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
