# Peer review of "“Talking on the Phone Is Very Cold”—Primary Health Care Nurses’ Approach to Enabling Patient Participation in the Context of Chronic Diseases during the COVID-19 Pandemic"

_healthcare, 2022, doi:10.3390/healthcare10122436_

Round 1

Reviewer 1 Report

 This manuscript presents the original research in this field. Generally, I consider this manuscript useful for the field of healthcare.

However, the key limitation of the study is the small number of the examinees. Thus, it is needed to mention the crucial limitations of the study.

Also, there are few suggestions:

- ABSTRACT: I suggest to the authors to add some specific results obtained by this study, for example some numerical data (%) which express prominent findings and results.

- METHODS AND MATERIALS: There is a lot of data, and it is somewhat difficult to follow the text for the reader. So I suggest to authors to shorten the text (where possible).

-Sentences in italic „….“ could be written as short descriptions – I think that these sentences could be moved, for instance, to one table which can present various examined factors (or to supplement material) and the text would be more clearly presented in that case. Also, I suggest the authors add the additional table (or 2 tables) which presents each steps of methods during the study periods (e.g. graphic presentation) + short conclusions for each theme mentioned/written in details from italic parts of the manuscript.

-If possible, authors could add a few sentences which describe specific COVID-19 conditions and public measures in this country during the specific periods of research. It is needed to mention public restrictions which limited healthy care during two mentioned times of research. What about quarantene?

-DISCUSSION: Please add more results from other research and compare your results with other authors' results.

-References are written with different types of initial letters i.e. there are mixed capitals and small letters (uppercase and lowercase letters).

 (FEW MINOR REVISIONS ARE NEEDED)

Author Response

please find the answers to your comments in the document attached

Reviewer 2 Report

The paper is an example of a solid work of qualitative method. It is especially appreciated that the authors present their work using a standard reporting guideline (i.e., CONSORT). However, for the paper to be finally ready to be published some issues need to be addressed.

Originality and novelty

The paper describes, and cites, several studies that report effect of the COVID-19 pandemic on PHC practices. A more elaborate explanation of why, and how, this paper contributes to knowledge would be advised (e.g., line 97-100).

Aim and research questions

The aim of the study is to “investigate how PHC nurses changed their approach to support the participation of patients”. The concept of approach needs to be clarified.

Also, the aim is somewhat differently described in the introduction and in the discussion (line 397-399). The aim could either be (1) to identify how PHC nurses approach changed during the pandemic, or (2) to investigate what PHC nurses did during the pandemic. It is not entirely clear which of these aims the paper aims to investigate. If it is change you are after, then you need to fully describe the situation before the pandemic and make a comparison.

The research questions: is “perceive” really the word you are looking for? Is it not more of a description of reported changes in practical behaviour due to the COVID-19 pandemic?

Discussion

Results regarding research question 3 is found only in the discussion section (line 460-483). This should be moved and presented in the results.

It would be better to move the sentences describing future research and gather them in one coherent section at the end of the discussion.

Strengths and limitations. Please explain and elaborate on how these limitations were addressed. It is not enough to list the limitations, you also need to discuss them (e.g., line 515)!

Author Response

Please find the answers to your comments in the document attaches

Author Response

Please find the answers to your comments in the document attached

Round 2

Reviewer 1 Report

The authors accepted the recommendations and answered on the questions.

Reviewer 3 Report

Thank you